# Uterine Fibroid Prevalence in a Predominantly Black, Chicago-Based Cohort

**DOI:** 10.3390/ijerph21020222

**Published:** 2024-02-14

**Authors:** Sithembinkosi Ndebele, Tecora Turner, Chuanhong Liao, Briseis Aschebrook-Kilfoy, Nina Randorf, Habibul Ahsan, Kunle Odunsi, Obianuju Sandra Madueke-Laveaux

**Affiliations:** 1Department of Public Health Sciences, The University of Chicago, Chicago, IL 60637, USA; sndebele@uchicago.edu (S.N.); cliao@bsd.uchicago.edu (C.L.); bkilfoy@bsd.uchicago.edu (B.A.-K.); randorf@bsd.uchicago.edu (N.R.); hahsan@bsd.uchicago.edu (H.A.); 2Pritzker School of Medicine, The University of Chicago, Chicago, IL 60637, USA; tecora@uchicago.edu; 3Institute for Population and Precision Health, The University of Chicago, IL 60637, USA; 4Comprehensive Cancer Center, The University of Chicago, Chicago IL 60637, USA; odunsia@bsd.uchicago.edu; 5Department of Obstetrics and Gynecology, University of Chicago, Chicago, IL 60637, USA

**Keywords:** uterine fibroids, leiomyomas, myomas, fibroids, pollution, environmental justice

## Abstract

(1) Objectives: To investigate the effect of individual-level, neighborhood, and environmental variables on uterine fibroid (UF) prevalence in a Chicago-based cohort. (2) Methods: Data from the Chicago Multiethnic Prevention and Surveillance Study (COMPASS) were analyzed. Individual-level variables were obtained from questionnaires, neighborhood variables from the Chicago Health Atlas, and environmental variables from NASA satellite ambient air exposure levels. The Shapiro–Wilk test, logistic regression models, and Spearman’s correlations were used to evaluate the association of variables to UF diagnosis. (3) Results: We analyzed 602 participants (mean age: 50.3 ± 12.3) who responded to a question about UF diagnosis. More Black than White participants had a UF diagnosis (OR, 1.32; 95% CI, 0.62–2.79). We observed non-significant trends between individual-level and neighborhood variables and UF diagnosis. Ambient air pollutants, PM2.5, and DSLPM were protective against UF diagnosis (OR 0.20, CI: 0.04–0.97: OR 0.33, CI: 0.13–0.87). (4) Conclusions: Associations observed within a sample in a specific geographic area may not be generalizable and must be interpreted cautiously.

## 1. Introduction

Uterine fibroids (UFs) are the most common benign neoplasm affecting women of reproductive age [1]. They are the leading cause of hysterectomy in the US and worldwide and are a source of significant socioeconomic burdens [1]. Black women are disproportionately affected by UFs, with a higher disease prevalence, earlier onset of disease, and more severe symptoms and disease progression [2]. This disproportionate burden of UFs and other female health conditions is increasingly understood in a framework of health inequity and the social and structural drivers of health [3]. Well-established risk factors that may contribute to the high prevalence of UFs in Black individuals include socioeconomic status, adverse environmental exposures, and experiences that increase chronic stress [4,5]. Each of these factors is believed to converge to increase inflammation within the uterine myometrium, resulting in somatic mutations (such as Med12) that transform normal myometrium stem cells and lead to UF tumor formation [6].

In addition, many lifestyle and socioeconomic factors, such as BMI, alcohol use, income, and occupation, correlate closely with neighborhood characteristics, e.g., access to healthy food and healthcare, exposure to environmental pollutants, and concentrated poverty [5,6]. Neighborhood poverty has been widely studied and is identified as a possible determinant of UF prevalence [7,8]. Poor and disenfranchised neighborhoods are often characterized by high crime rates, food insecurity, and other important social determinants of health [9,10,11]. Lastly, while respiratory and cardiovascular diseases are most often linked to air pollution, recent studies have shown that air pollution is positively associated with the risk of gynecological diseases [12,13], and exposure to air pollutants such as ozone and PM 2.5 may contribute to the racial disparities in UF incidence, prevalence, and severity [14]. The biological mechanism by which air pollutants, e.g., ozone, increase fibroid formation is unclear. Theories such as oxidative stress and immune-inflammatory and hypertension-mediated pathways have been proposed [13].

Cook County, of which 85% is Chicago, has been ranked among the worst 10% of counties in the United States air quality indicators [15]. Therefore, Chicago provides a unique opportunity to examine the potential impact of air pollutants, as well as other urban risk factors, on the prevalence of UFs. Since 2013, a predominantly Black population on Chicago’s South Side has been enrolled in the Chicago Multiethnic Prevention and Surveillance Study (COMPASS) with the goal of mitigating health disparities [16]. To this end, extensive data have been collected to understand individual, neighborhood, and environmental factors relevant to disease prevention, disparity mitigation, and improved health outcomes. In this study, we analyzed data from a sample of participants enrolled in this existing longitudinal cohort study to assess the relationship between individual-level, neighborhood, and environmental variables and UF prevalence.

## 2. Methods

### 2.1. Study Design

This cross-sectional study analyzed the baseline data of a sample of participants from COMPASS. Data included in this study were collected from July 2019–May 2020. A more detailed description of the COMPASS study design can be found elsewhere [16].

### 2.2. Study Population

The sample analyzed in this study was obtained from COMPASS. Established in 2013, COMPASS is a longitudinal prospective cohort study that includes 8000 participants in the City of Chicago. Its purpose is to assess the influence of factors, such as neighborhood, environment, exposure to air pollutants, socioeconomic status, healthcare access, lifestyle, behavior, and genetics, on the health of Chicagoans. COMPASS enrolls residents of the greater Chicago area who are at least 18 years of age and not incarcerated at the time of enrollment. The survey was designed by co-authors Drs. Aschebrook-Kilfoy and Ahsan. Most survey items are harmonized with existing NIH/NCI surveys.

To investigate the possible correlation between these above-mentioned factors and UF diagnosis, we analyzed data of COMPASS participants who responded to the question, “Has a doctor or healthcare professional ever diagnosed you with uterine fibroids?” Based on their responses, we categorized participants into two groups: those who had received a UF diagnosis (yes) and those who had not (no). 

### 2.3. Individual-Level Variables

Demographic factors such as age, race, and ethnicity, as well as lifestyle and behavioral factors, including activity levels, alcohol intake, and smoking, were reported through the questionnaire. Additionally, access to healthcare, neighborhood factors, such as crime and safety, socioeconomic status, including employment status and income status, and reproductive history, including pregnancy and hysterectomy, were reported through the questionnaire. All participants in the sample listed female as their gender. We categorized participants as either active or inactive based on their reported participation in at least one of the 15 physical activities listed in the questionnaire (ranging from household chores to vigorous workouts). Participants were classified as “smokers” if they reported smoking cigarettes, cigars, marijuana, or vaping nicotine and/or tobacco daily or weekly. Participants were classified as “alcohol consumers” if they reported regular alcohol consumption and the intake of multiple alcoholic beverages within the last 12 months. Employment status was divided into four categories: employed, unemployed, retired, or unknown. Income status was divided into three categories: low income (USD 34,999 or less), middle income (USD 35,000–USD 89,999), and high income (USD 90,000 or above).

Access to healthcare was assessed using two variables: “access to care” and “quality of care”. The metric for access to care was determined by combining participants’ responses to questions on where they go for health care (i.e., urgent care, emergency room, clinic visit, etc.), their perception of the number of doctors in their community, and whether they had ever been turned away by a doctor for financial or insurance reasons. Quality of care was evaluated based on participants’ satisfaction with the care they received in the last 12 months and their agreement with statements about their doctors’ medical knowledge and the amount of time spent with patients.

### 2.4. Neighborhood Variables

To investigate the possible impacts of participants’ perception of neighborhood crime and violence on physical activity levels, participants’ responses to questions regarding their choice to forego exercise due to concerns about crime and violence, as well as the impact of these concerns on their daily lives, were assessed. Contextual neighborhood variables were analyzed using Chicago Health Atlas (CHA) data for each Chicago community area between 2015 and 2019 [17]. Chicago has distinct community areas (aka neighborhoods). COMPASS links survey data to community areas, and CHA data are merged into COMPASS data on the shared community area level. Six neighborhood variables were included: the hardship index (composite score reflecting hardship in the community), the neighborhood safety rate (% of adults who report that they feel safe in their neighborhood “all the time” or “most of the time”), low food access (% of residents who must travel further than ½ mile to the nearest supermarket in urban areas or 10 miles in rural areas), traffic intensity (proximity to vehicle traffic), the social vulnerability index (percentile relative vulnerability based on social factors), and the rate of received needed care (% of adults who report that it is “usually” or “always” easy to obtain care with their health plan).

### 2.5. Environmental Variables

Ambient exposure data, including PM2.5, ozone, diesel particulate matter (DSLPM), and proximity to traffic (PTRAF), was extracted from COMPASS, which obtains air quality data by geocoding participant-supplied addresses and linking them to one of 77 Chicago community areas and their census tract or block. These ambient exposure levels are derived from the 2019 Environmental Justice Screening (EJSCREEN) air quality data and merged with the COMPASS data set using the EJSCREEN ID variable at the census Federal Information Processing Standards (FIPS) code block group level. 

### 2.6. Statistical Analysis

Descriptive statistics of the investigated variables from the COMPASS dataset and the selected contextual neighborhood variables retrieved from the Chicago Health Atlas (CHA) for each Chicago community were linked to individual participants. The mean ± standard deviation (SD) or median [interquartile range (IQR)] was reported for continuous variables based on data normal distribution, and frequencies and percentages were presented for categorical variables. The Shapiro–Wilk normality test was used to examine whether the variables were normally distributed. Differences in subject characteristics between groups were analyzed by two-sample t-tests or Mann–Whitney tests, depending on the distributions for continuous variables, and by Pearson’s chi-squared or Fisher’s exact test for categorical variables. Logistic regression was used to investigate trends in age, race, access to quality care, behavioral lifestyle, contextual neighborhood factors, socioeconomic status, and ambient exposures related to UF, illustrating the odds ratio (OR) value with a 95% confidence interval (CI). In addition, potential risk factors for UFs were identified in the multivariable logistic regression model. Of note, a multilevel model was not utilized to analyze the contextual neighborhood variables because the available data had no hierarchical or clustered structure. We used Spearman’s rank correlation coefficient with Bonferroni correction to assess the contextual neighborhood correlations because the contextual neighborhood data did not have approximately normal distributions. Mixed positive and negative correlations did not satisfy the critical assumption of unidirectionality needed for the weighted quantile sum (WQS) analysis for the overall mixture effect of neighborhood characteristics, which was performed in a similar study by a co-author, Dr Aschebrook-Kilfoy [18,19]. Lastly, multivariable logistic regression was performed to assess the impact of contextual neighborhood variables on UF diagnosis adjusted by race and household income status. Two-sided *p* < 0.05 was considered statistically significant. All the analyses were conducted using Stata/SE software 17.0 (StataCorp LLC, College Station, TX, USA).

## 3. Results

A total of 602 participants aged 35–76 years (mean (SD): 50.3 ± 12.3) met the criteria for this study, and 21% self-reported a UF diagnosis. See Table 1 for a summary of participants’ demographics, lifestyle, and reproductive history. Univariate analysis of each variable is reported in this section unless otherwise indicated.

### 3.1. Individual-Level Variables

#### 3.1.1. Demographic Factors

The average age of participants with a UF diagnosis was 37 years. In our sample, 85% identified as Black, 9% as White, and 6% as other. The odds of a UF diagnosis decreased with age (OR, 0.85; 95% CI, 0.83 to 0.88). Black participants had higher odds of a UF diagnosis when compared to White participants (OR, 1.32; 95% CI, 0.62 to 2.79) or others (OR, 1.11; 95% CI, 0.37 to 3.32). A total of 90.7% of the sample were non-Hispanic, 7.6% were unknown, and 1.7% were Hispanic. Participants of Hispanic ethnicity had lower odds of a UF diagnosis (OR, 0.43; 95% CI, 0.05 to 3.40). The mean age of participants with a UF diagnosis was lower in Black (36.5 years) compared to White (41.6 years) or participants of other races (41 years), with a *p*-value of 0.330 (Figure 1A).

#### 3.1.2. Socioeconomic Factors

Seventy percent of the participants were in the low-income bracket. Those in higher income brackets had increased odds of a UF diagnosis. Unemployed participants had decreased odds of a UF diagnosis (OR, 0.79; 95% CI, 0.48 to 1.28), and 39% of the sample participants reported having no access to quality healthcare. Patients with access to quality care were approximately 26% less likely to receive a UF diagnosis (OR, 0.74; 95% CI, 0.50 to 1.09). A total of 42% of the sample participants reported having an insufficient number of doctors in their community. Participants who reported having enough doctors in their community had lower odds of a UF diagnosis (OR, 0.96; 95% CI, 0.61 to 1.52). A total of 42% of participants reported concerns about crime and neighborhood violence. Participants who had daily concerns about crime trended towards higher odds of receiving a UF diagnosis (OR, 1.19; 95% CI, 0.79 to 1.79) compared to those who did not have these concerns (Figure 1B).

#### 3.1.3. Lifestyle and Behavioral Factors

Fifty-two percent of participants were categorized as obese (BMI > 30), 24% as overweight (BMI 25 to <30), 20% as healthy weight (BMI 18.5 to <25), and 4% as underweight (BMI < 18.5). We observed a positive trend between BMI and a UF diagnosis; obese participants had 1.5 times higher odds of a UF diagnosis compared to participants with a normal BMI (OR, 1.49; 95% CI, 0.87 to 2.56). UF diagnoses were less likely in participants who reported daily exercise (OR, 0.93; 95% CI, 0.59 to 1.48). Within our sample, 88.5% of the participants reported having an active lifestyle, and these participants were more likely to have a UF diagnosis (OR, 1.47; 95% CI, 0.75 to 2.89). Fifty-two percent of the sample participants were smokers, and they were 1.28 times more likely to have a UF diagnosis (OR, 1.28; 95% CI, 0.86 to 1.90). Additionally, 44% of the sample participants reported childhood secondhand smoke exposure, which was associated with increased odds of a UF diagnosis (OR, 1.46; 95% CI, 0.84 to 2.52). Of the sample participants, 17% reported regular alcohol consumption (every day or every week), 40% denied regular alcohol use, and 42% reported unknown alcohol usage. Those who reported regular alcohol use were 1.14 times more likely to have a UF diagnosis (OR, 1.14; 95% CI, 0.68 to 1.91) (Figure 1C).

#### 3.1.4. Reproductive History

A history of pregnancy was reported by 81% of participants. These participants had higher odds of a UF diagnosis (OR, 1.42; 95% CI, 0.83 to 2.43). The odds of a UF diagnosis were also higher in participants who experienced pregnancy loss (OR, 1.42; 95% CI, 0.73 to 2.75). Abortions were slightly more common among participants with a UF diagnosis (OR, 1.19; 95% CI, 0.43 to 1.30). Sixteen percent of participants reported having a hysterectomy, while 84% denied having undergone the procedure. The odds of a UF diagnosis were 8.9 times higher in participants who had undergone a hysterectomy (OR, 8.91; 95% CI, 5.52 to 14.37) (Figure 1D).

### 3.2. Neighborhood Variables

Except for traffic intensity, neighborhood contextual characteristics were similar across groups (Table 2). Each selected neighborhood characteristic showed no significant association with a UF diagnosis (Figure 1F). Spearman’s correlation of the six neighborhood characteristics showed moderate correlations between several. Spearman’s correlation coefficient indicated a value of 0.82 between the hardship index and social vulnerability and 0.63 between the hardship index and neighborhood safety (Figure 2). When stratified by individual-level variables, race, and household income status, the six neighborhood characteristics did not have a statistically significant impact on the odds of a UF diagnosis. When adjusted for age, traffic intensity was slightly protective against a UF diagnosis.

### 3.3. Environmental Variables

PM2.5 was associated with decreased odds of a UF diagnosis (OR, 0.20; 95% CI, 0.04 to 0.97). DSLPM exposure decreased the odds of UF diagnosis (OR, 0.33; 95% CI, 0.13 to 0.87). Ozone levels did not follow a normal distribution, and median concentration exposures were similar in both groups at 45.409 μg/m^3^. Additionally, ozone exposure decreased the odds of a UF diagnosis, although the effect was not statistically significant (OR, 0.69; 95% CI, 0.33 to 1.41). Average PTRAF exposure was 9.825 μg/m^3^, and it did not significantly impact the odds of a UF diagnosis in our sample (Figure 1E). Of note, other multivariable analyses performed did not meet the Hosmer–Lemeshow goodness of fit test.

## 4. Discussion

In recent years, considerable attention has been given to understanding the role of social, economic, and environmental factors on health inequity [20]. This study explores UF prevalence among predominantly Black urban residents in Chicago, considering individual, neighborhood, and environmental factors. Chicago’s unique features—socioeconomic profile, demographic composition, high traffic, and industrial presence, leading to poor air quality—make it an ideal location for assessing UF prevalence [15]. However, these features affect the generalizability of our findings. Furthermore, the small sample size, relatively narrow exposure distribution, and oversampling of non-Hispanic Blacks, who are disproportionately impacted by UFs, may explain the lack of statistically significant findings. 

In this study, 85% of participants were non-Hispanic Blacks. Black participants had a higher likelihood of a UF diagnosis, and we observed a positive correlation between a UF diagnosis and lifestyle and demographic factors such as regular alcohol use, secondhand smoke exposure, elevated BMI, and infrequent exercise. These findings are consistent with well-established data [1,3,5,9]. The association we found between an active lifestyle and higher odds of a UF diagnosis was unexpected and may be attributable to physical activity overestimation bias since our classification process was based on participants’ self-report of engagement in activities [21]. Forty-two percent of participants reported concerns about crime and violence, which impacted their ability to engage in outdoor physical activity and affected their daily lives. Participants with these concerns had higher odds of a UF diagnosis. Despite the lack of statistical significance, which could be attributed to an overall small sample size, this correlation is expected because crime and violence are a significant source of chronic stress, which can lead to allostatic load and a subsequent pro-inflammatory state [22]. Chronic inflammation has been implicated in the development of UFs and may be a critical contributing factor to the racial disparity observed in UF diagnoses. The increased odds of a UF diagnosis in participants reporting a history of pregnancy loss and hysterectomy are consistent with published data and underscore the substantial morbidity associated with UFs, as well as their negative impact on quality of life [1,23]. Although research has suggested that pregnancy protects against UF occurrence [24], our study found that participants who had been pregnant before had higher odds of a UF diagnosis. This finding could be due to the common occurrence of UF diagnoses during pregnancy.

Neighborhood characteristics, independently and stratified by individual-level variables (race and household income), did not significantly influence UF diagnoses. This was an unexpected finding, and we suspect it is due to an overall low sample size, i.e., Type II error [25]. Furthermore, no statistically significant correlation was found between ozone or PTRAF exposure and UF diagnoses, whereas DSLPM and PM2.5 showed statistically significant negative correlations with UF diagnosis. These findings were not as expected because previous studies have explored the association between air pollutants and UFs, with some reporting a modest increased risk of UF with ozone and PM2.5 exposure [12,13,14]. Our findings may differ from prior studies due to overall lower levels and narrower ranges of ozone (44–46 μg/m^3^ vs. 50.74–71.04 μg/m^3^ in Black Woman’s Health Study) and average PM2.5 (9.82 μg/m^3^ vs. 13.6 μg/m^3^ in Black Woman’s Health Study and 15.3 μg/m^3^ in The Nurses’ Health Study II). Although our findings do not invalidate previous data, they suggest a possible threshold exposure level where UF risk increases, and larger variations in exposure levels may allow differences to be observed, while smaller variations reduce the ability to detect such differences. 

### 4.1. Strengths and Limitations

This study has several strengths. First, the disadvantaged group most impacted by UF is adequately represented in our study sample. Second, our data source, COMPASS, provides access to specific neighborhood characteristics using participant-supplied addresses instead of proxy variables. Third, this paper investigates an important and understudied issue of the social and environmental causes of health inequity in UF prevalence. There are several limitations of this study. First, despite access to a large cohort, we performed a cross-sectional analysis of a relatively small sample of 602 participants who completed the survey module assessing UF diagnosis. The analyzed sample was not nationally or geographically representative, and although the over-representation of disadvantaged groups is a strength, it limits the generalizability of our study findings. Second, the age and report of UF diagnosis are not validated by medical data. This, along with other findings, may be influenced by questionnaire and recall bias [26]. Although validation of fibroid diagnosis is preferred, previous data suggests that patient-report is accurate for over 90% of patients with UF [27,28]. To mitigate these limitations for future studies using COMPASS, we will request that a UF diagnosis be included in all surveys, as well as questions addressing age at diagnosis and verification of an image-confirmed diagnosis. Lastly, the lack of a temporal association between air pollutant data collection and the date of UF diagnosis, due to the nature of survey response collection, limits the interpretation of our findings on the impact of air pollution exposure on a UF diagnosis. Future studies with larger sample sizes, wider exposure distributions, inclusion of medical record data, and more comprehensive data collection methods will contribute to a deeper understanding of the factors influencing UF diagnoses.

### 4.2. Further Research

To reflect the association more accurately between exposures and disease diagnosis, studies evaluating the impact of socioeconomic, lifestyle, and environmental factors on UFs should capture both the age at diagnosis and the duration of environmental exposures leading to or at the time of diagnosis. Additionally, participants from cohorts specifically designed to investigate health conditions, such as UFs, should be sampled for analysis. Alternatively, existing cohorts, such as COMPASS, could be modified accordingly to avoid limited and inaccurate information about the condition in question. Lastly, similar studies using population-based cohorts could enhance heterogeneity and variability within the cohort, therefore improving the generalizability of the study results. 

## 5. Conclusions

The impact of structural and environmental factors on UF development is a growing area of research interest. Our investigation of this relationship in a predominantly Black Chicago-based cohort, which includes individuals residing in historically disenfranchised communities of South Chicago, did not reveal significant associations between these structural drivers and UF prevalence. However, our study provides foundational insights into the cohort that we queried and identifies an opportunity to leverage an existing longitudinal cohort study by expanding its variables to include gynecologic-specific data that would improve the robustness of future analysis. Future analysis with more robust data may allow us to determine if there is a significant association between structural and environmental variables and UF prevalence. Identifying this relationship, if it exists, would provide a justifiable platform to pursue policy changes. 

## Figures and Tables

**Figure 1 ijerph-21-00222-f001:**
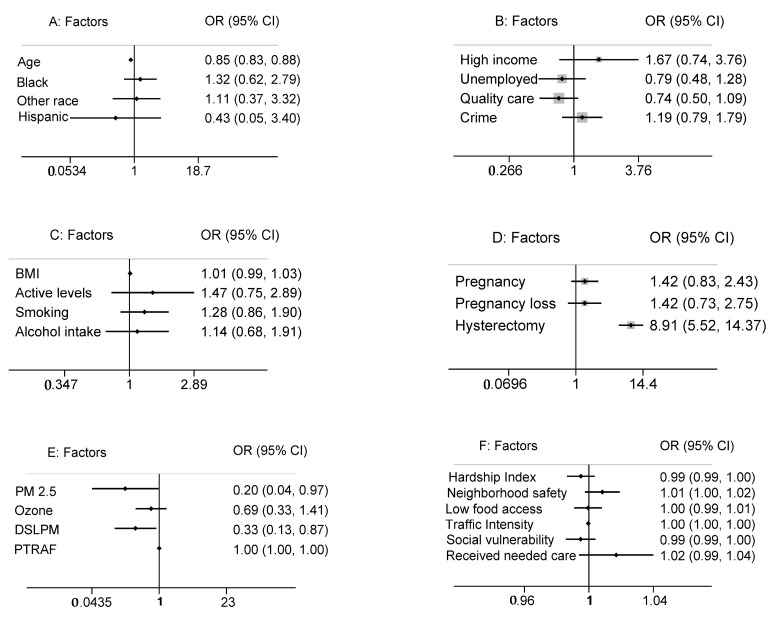
(**A**) **Odds of UF diagnosis by Demographic factors: Age, Race (Black), ethnicity.** Odds ratio and 95% Confidence Interval (CI) were utilized for an interquartile range where continuous predictors of age compared quartile 3 with quartile 1. Categorical predictors of Race, Daily Exercise, and Active Lifestyle utilized simple odds and compared them to a reference group (White, normal BMI, No exercise, not active lifestyle). Odd ratios above 1 indicate an increased risk of developing uterine fibroid. Age: continuous variable. Race: White, Black, and Other. (**B**) **Odds of UF diagnosis by Income/Employment Status, Access to Quality Care, and Crime.** Categorical predictors of Income status, Employment status, access to quality care, and crime utilized simple odds and compared them to a reference group (low income, employed, no quality care, not enough doctors, and no crime). Odd ratios above 1 indicate an increased risk of developing uterine fibroid. Conversely, an odds ratio of less than 1 represents a protective effect. (**C**) **Odds of UF diagnosis by lifestyle and behavioral: BMI, activity levels, alcohol intake, smoking.** Odds ratio and 95% Confidence Interval (CI) were utilized for an Interquartile range where continuous predictor BMI compared quartile 3 with quartile 1. Categorical predictors of smoking status, secondhand smoking exposure, and alcohol consumption utilized simple odds and compared them to a reference group (normal BMI, No exercise, not active lifestyle, no alcohol intake, and no smoking). Odd ratios above 1 indicate an increased risk of uterine fibroid diagnosis. BMI: Underweight, Normal, Overweight, Obese. Daily exercise: No exercise, daily exercise. Active lifestyle: Inactive lifestyle, active lifestyle. (**D**) **Odds of UF diagnosis by pregnancy and hysterectomy history.** Categorical predictors’ previous pregnancy status, Pregnancy loss experience, and hysterectomy utilized simple odds and compared them to a reference group of zero or not experienced. Odd ratios above 1 indicate an increased odds of uterine fibroid diagnosis. (**E**) **Odds of UF diagnosis by Ambient Exposure Ranges.** Odd ratios above 1 indicate an increased risk of developing uterine fibroid. Conversely, an odds ratio of less than 1 represents a protective effect. (**F**) **Odds of UF diagnosis by Neighborhood contextual variables.** Odd ratios above 1 indicate an increased risk of developing uterine fibroid. Conversely, an odds ratio of less than 1 represents a protective effect.

**Figure 2 ijerph-21-00222-f002:**
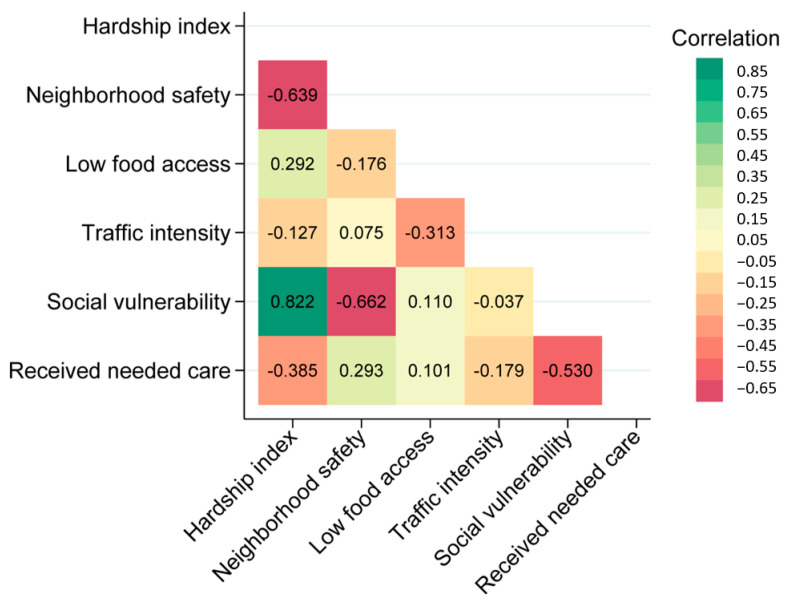
**Spearman correlation of Neighborhood contextual variables.** Neighborhood contextual factors are categorized into interquartile ranges. Median hardship index (score), neighborhood safety rate (% of adults), low food access (% of residents), traffic intensity (distance-weighted vehicles), social vulnerability index, and received needed care rate (% of adults).

**Table 1 ijerph-21-00222-t001:** Summary of Participants’ demographics, lifestyle, and reproductive history.

	Entire Sample (n = 602)	Fibroid Diagnosis (n = 127)	No Fibroid Diagnosis (n = 475)	*p*-Value
**Age (year)**, mean ± SD	50.3 ± 12.3	37.1 ± 10.5	53.8 ± 10.1	<0.001
**BMI**, mean ± SD	31.4 ± 9.0	32.2 ± 7.8	31.2 ± 9.3	0.265
**Race**, n (%)				
Black	513 (85.2)	111 (87.4)	402 (84.6)	0.792
White	52 (8.6)	9 (7.1)	43 (9.1)	
Other	37 (6.2)	7 (5.5)	30 (6.3)	
**Ethnicity**, n (%)				
Non-Hispanic	546 (90.7)	113 (89.0)	433 (91.2)	0.391
Hispanic	10 (1.7)	1 (0.8)	9 (1.9)	
Unknown	46 (7.6)	13 (10.2)	33 (7.0)	
**Socioeconomic Status**, n (%)				
*Employment Status*				
Employment	127 (21.1)	30 (23.6)	97 (20.4)	0.360
Unemployed	368 (61.1)	72 (56.7)	296 (62.3)	
Retired	65 (10.8)	18 (14.2)	47 (9.9)	
Unknown	42 (7.0)	7 (5.5)	35 (7.4)	
*Income Status ^a^*				
Low income	422 (70.1)	83 (65.4)	339 (71.4)	0.337
Middle income	44 (7.3)	8 (6.3)	36 (7.6)	
High Income	31 (5.2)	9 (7.1)	22 (4.6)	
Unknown	105 (17.4)	27 (21.3)	78 (16.4)	
**Behavioral Lifestyle**, n (%)				
*Alcohol/Smoking Status*				
*Smoking Status*				
Smoker	312 (51.8)	72 (56.7)	240 (50.5)	0.231
Non-Smoker	290 (48.2)	55 (43.3)	235 (49.5)	
*Alcohol Consumption*				
Consumer	105 (17.4)	29 (22.8)	76 (16.0)	0.623
Non-Consumer	243 (40.4)	61 (48.0)	182 (38.3)	
Unknown	254 (42.2)	37 (29.1)	217 (46.7)	
**Reproductive History**, n (%)				
*Pregnancy outcome*				
Live birth	375 (62.3)	79 (62.2)	296 (62.3)	0.385
Pregnancy loss	51 (8.47)	14 (11)	37 (7.8)	
Abortion	62 (10.3)	15 (11.8)	47 (9.9)	
Not reported	114 (18.9)	19 (15)	95 (20)	
*Hysterectomy*	97 (16.4)	57 (45.6)	40 (8.6)	<0.001

Income Status ^a^: Income status ranges. Low income: USD 34,999 or less. Middle income: USD 35,000–USD 89,999. High income: USD 90,000 or above.

**Table 2 ijerph-21-00222-t002:** Summary of Neighborhood Characteristics.

Neighborhood Characteristic, Median (Interquartile Range)				
	Entire Sample (n = 602)	Fibroid Diagnosis (n = 127)	No Fibroid Diagnosis (n = 475)	*p*-Value
Hardship Index	83.1 (75.3–89.3)	83.1 (57.3–86.9)	83.1 (75.3–89.8)	0.338
Neighborhood safety	47.1 (33.6–58.2)	47.5 (36.5–59.2)	46.3 (33.6–58.2)	0.130
Low food access	36.9 (22.3–63.5)	36.9 (22.3–63.5)	36.9 (22.3–63.5)	0.768
Traffic Intensity	615.3 (411.5–1630.2)	563.1 (411.5–1013.0)	619.2 (411.5–1630.2)	0.031
Social vulnerability	81.5 (72.7–83.5)	81.5 (69.4–82.6)	81.5 (74.0–83.5)	0.284
Received needed care	77.2 (72.7–86.5)	79.3 (74.6–87.6)	77.2 (72.7–86.2)	0.143

## Data Availability

Data are available upon reasonable request. COMPASS will continue to collect a rich set of data on multiple exposure domains and health outcomes. For more information, refer to the website compass.uchicago.edu (accessed on 2 December 2023). Researchers interested in collaboration are invited to propose research questions based on the data available within COMPASS or to submit a request for additional data collection. Requests can be submitted electronically on the COMPASS website and will be reviewed by the COMPASS scientific board. The COMPASS study team is particularly interested in collaborations that will enhance research methods for this type of work, assess the impact of environmental exposure, highlight exposures of key significance in urban communities, and address health issues of concern in Chicago and other urban centers.

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
