# Peer review of "Uterine Fibroid Prevalence in a Predominantly Black, Chicago-Based Cohort"

_ijerph, 2024, doi:10.3390/ijerph21020222_

Round 1

Reviewer 1 Report

Comments and Suggestions for Authors

I appreciate the opportunity to review the manuscript entitled “Uterine Fibroid Prevalence in a predominantly Black, Chicago-Based Cohort” submitted to the journal IJERPH.  

The authors

Reviewer Comments:

1.      Please prepare a list of abbreviations used in the manuscript.

2.      Please mention the role of the pollutants in gynecological / pregnancy-related diseases, especially uterine leiomyoma.

3.      There are just 26 references cited in the manuscript….Please make the number of references above 30 (check the journal guidelines).

Taking into account the significance of the role of the pollutants in uterine leiomyoma pathogenesis, my opinion is that this submission meets the criteria to be published in the journal IJERPH after major revisions and inclusion of the data I suggested.

Reviewer 2 Report

Comments and Suggestions for Authors

Thank you for allowing me to review this article.

The authors have essentially conducted a sub-analysis of the COMPASS study addressing the prevalence of uterine fibroids in a predominantly black female cohort from Chicago.

The information obtained was derived from a questionnaire determined by the authors and sent out to the incumbents of the COMPASS cohort, which had been recruited from 2013. The questionnaire  was completed between mid 2019-2020. The women were identified as having uterine fibroids based on the question of " whether the participant had ever been told by a doctor that she had fibroids ".

The study has been well written with no obvious spelling or grammatical errors.

The authors need to ;

(1) Expand on why all the variable factors were considered, specifically factors such as ozone, traffic intensity, other weather variables etc.

(2) The impact of only 609 women replied out of 8000 incumbents of the study and why were women included unto 79 years of age ?

(3) If there was no clinical assessment to confirm the diagnosis of uterine fibroids at recruitment, subsequently or after the questionnaire, what proof is there that the fibroids existed.

(4) No information is provided pertaining to time exposed to the variables analysed. Was the variable considered impacting prevalence of fibroids from date of entry to COMPASS, from the time when the patient was told she had fibroids or from time of filling in the questionnaire.

Reviewer 3 Report

Comments and Suggestions for Authors

There are some major concerns in the revised paper

- what is the rationale of presentation study after 10 years, currently statistics probably might be different.

- PRISMA flow diagram was not shown so we don't know what were the inclusion and exclusion criteria... -> 8.000 participants -> 602 entire sample.

- low level of novelty, what was the aim of the study? There are in general many already known facts.

- discussion in such well-know area should be expanded and much more comprehensive.

- references should be refreshed and expanded.

Round 2

Reviewer 3 Report

Comments and Suggestions for Authors

I accept all responses and clarifications given by the authors. At present I can recommend paper for further steps of editorial process and I accept it in its present form.